# Physical Activity in Adults with Schizophrenia and Bipolar Disorder: A Large Cross-Sectional Survey Exploring Patterns, Preferences, Barriers, and Motivating Factors

**DOI:** 10.3390/ijerph20032548

**Published:** 2023-01-31

**Authors:** Garry A. Tew, Laura Bailey, Rebecca J. Beeken, Cindy Cooper, Robert Copeland, Samantha Brady, Paul Heron, Andrew Hill, Ellen Lee, Panagiotis Spanakis, Brendon Stubbs, Gemma Traviss-Turner, Lauren Walker, Stephen Walters, Simon Gilbody, Emily Peckham

**Affiliations:** 1Institute for Health and Care Improvement, York St John University, York YO31 7EX, UK; 2School of Medicine, University of Leeds, Leeds LS2 9JT, UK; 3School of Health and Related Research, University of Sheffield, Sheffield S1 4DA, UK; 4Advanced Wellbeing Research Centre, Sheffield Hallam University, Sheffield S9 3TU, UK; 5Department of Health Sciences, University of York, York YO10 5DD, UK; 6School of Psychology, Mediterranean College, 104 34 Athens, Greece; 7Institute of Psychiatry, Psychology and Neuroscience, King’s College London, London WC2R 2LS, UK

**Keywords:** exercise, sedentary behaviour, cross-sectional study, survey, determinants, preferences, severe mental illness

## Abstract

Adults with severe mental ill health may have specific attitudes toward physical activity. To inform intervention development, we conducted a survey to assess the physical activity patterns, preferences, barriers, and motivations of adults with severe mental ill health living in the community. Data were summarised using descriptive statistics, and logistic regressions were used to explore relationships between physical activity status and participant characteristics. Five-hundred and twenty-nine participants (58% male, mean age 49.3 years) completed the survey. Large numbers were insufficiently active and excessively sedentary. Self-reported levels of physical activity below that recommended in national guidelines were associated with professional inactivity, consumption of fewer than five portions of fruit and vegetables per day, older age, and poor mental health. Participants indicated a preference for low-intensity activities and physical activity that they can do on their own, at their own time and pace, and close to home. The most commonly endorsed source of support was social support from family and friends. Common motivations included improving mental health, physical fitness, and energy levels. However, poor mental and physical health and being too tired were also common barriers. These findings can inform the development of physical activity interventions for this group of people.

## 1. Introduction

People with severe mental ill health (SMI) such as schizophrenia and bipolar disorder comprise between 2–4% of the population and have some of the worst health indices of any section of the UK population [1], experiencing a mortality gap of 15–20 years compared to people without SMI [2]. The main cause of these deaths is due to chronic physical conditions such as type 2 diabetes, cardiovascular, respiratory, and infectious diseases, and hypertension. People with SMI have a 2–3-fold increased risk of death from coronary heart disease compared with people without SMI [3]. They also have two times the risk of developing diabetes [4]. Whilst life expectancy in the general population has steadily increased over recent decades, life expectancy for people with SMI has declined [5]. This widening health inequality is a priority in the UK NHS Long Term Plan [6].

Sedentary behaviour (any waking behaviour characterised by an energy expenditure ≤1.5 metabolic equivalents while in a sitting, reclining, or lying posture) and physical inactivity (insufficient physical activity to meet physical activity guidelines) are leading causes of death worldwide [7]. The World Health Organization has stated that preventing sedentary behaviour and physical inactivity is as important as encouraging smoking cessation [8]. In the wider population, there is robust evidence that higher levels of physical activity and lower levels of sedentary behaviour can reduce cardiovascular disease, type 2 diabetes, and metabolic syndrome [9]. Previous small studies have suggested that physical activity can improve mental health symptoms in people with schizophrenia and major depression [10]. People with SMI however, experience several unique barriers to engaging in physical activity. These include increased mental health symptoms, lack of social support, and the side effects of medication [11]. A recent meta-analysis exploring sedentary behaviour in people with SMI established that people with SMI spent nearly 13 h per day being inactive, three hours per day more than people without SMI [12]. To support people with SMI to increase their physical activity and reduce sedentary behaviour, there is a need to understand the preferences for and determinants of physical activity in this group. Therefore, we conducted a large cross-sectional survey to investigate the following research questions:What are the patterns of physical activity and sedentary behaviour in people with SMI? (RQ1)What are the associations between physical activity status (meeting national physical activity guidelines or not) and selected sociodemographic and health variables in this population? (RQ2)What are the physical activity preferences of people with SMI? (RQ3)What barriers to participating in physical activity are reported by people with SMI? (RQ4)What motivating factors for participating in physical activity are reported by people with SMI? (RQ5)Do physical activity preferences, barriers, and motivating factors differ by gender, age group, physical activity status, and self-rated mental health in this population? (RQ6)

## 2. Materials and Methods

### 2.1. Ethics Approval

The study was approved by the West of Scotland Research Ethics Service in June 2018 (reference 18/WS/0107).

### 2.2. Study Design

This was a cross-sectional study in which a survey was completed online, via telephone, or post.

### 2.3. Participants

To be eligible to take part, participants had to be aged 18 years or over and have a documented diagnosis of schizophrenia, schizoaffective disorder, or bipolar disorder.

### 2.4. Recruitment and Procedures

The survey was conducted between December 2018 and March 2022. Prior to the COVID-19 pandemic, adults with SMI who had taken part in a previous study (Closing the Gap: Health and Well-being Cohort, described elsewhere [13]) and had consented to recontact, were sent a postal pack containing the survey, an invitation letter, a participant information sheet, and a freepost return envelope. The pre-pandemic recruitment method used mass mailouts involving five sites (three mental health trusts and two clinical commissioning groups). Recruitment was paused in March 2020 at the onset of the pandemic and recommenced in June 2021. Mass mailouts had become impractical as a recruitment method during the pandemic restrictions, due to the lack of physical access to office space and university postal systems being suspended, however, the research team had tested and had success with a mainly remote recruitment method. Participants were recruited from NHS Trust databases or through new clinical contacts and were contacted (in order of preference) by phone (to complete the questionnaire verbally with a researcher), online (being sent a link to an online version of the survey via email or SMS message) or hard copy by post, if other methods were unavailable or failed. This more open recruitment method, which was easier during the COVID-19 restrictions and more personal due to phone contact, was employed until the close of the study in March 2022. Voluntary completion of the survey indicated consent to participate in the study. The target sample size was 500 participants, based on the anticipated response rate and previous research [14].

### 2.5. Measures

A multi-disciplinary research team (comprising researchers and experts by experience) developed a bespoke survey, the full version of which was uploaded on the Open Science Framework (https://osf.io/rgc62/; accessed on 29 July 2022). It had three sections: (A) physical activity and sedentary behaviour; (B) health; (C) socio-demographics. It was expected to take about 20 min to complete. Respondents maintained anonymity. An overview of the measures included in the current study is provided below.

Participants’ physical activity and sedentary behaviours were assessed using items from the TAM2 questionnaire [15], the Health Survey for England [16], and the SIMPAQ [17]. The first two items of the TAM2 questionnaire were used to assess the volume of moderate to vigorous physical activity per week. Specifically, participants were asked how many times in an average week they engage in moderate and vigorous activities. They were also asked how many minutes they perform moderate and vigorous activities each time. The responses for frequency and duration were multiplied to determine the weekly volumes of moderate and vigorous activity. Participants were then classified according to their weekly volume of activity of at least moderate intensity: active (>150 min/week), fairly active (30–149 min/week), or inactive (<30 min/week) [18]. Minutes of vigorous-intensity activity were given twice the credit of minutes of moderate-intensity activity when combining moderate and vigorous intensity to calculate the equivalent combination [18]. Participants were also asked about the frequency and duration of muscle-strengthening activities and walking. Examples of muscle-strengthening activities were provided, based on those recognised by the Health Survey for England [16]. The weekly frequency of muscle-strengthening activity was summarised, and activities/sessions were only counted if carried out in bouts of 10 min or more. Participants’ weekly walking time was also summarised, and the previously mentioned activity classification was determined with and without the inclusion of walking time as moderate-intensity activity. Finally, the average time in bed per day and average sedentary time per day were assessed using the following two items from the SIMPAQ that were reworded for self-report: ‘In an average week, how much time do you spend in bed per night on average?’ and ‘Of the time you spend out of bed, how much time per day on average do you spend sitting or lying down, such as when you are eating, reading, watching TV or using electronic devices?’ [17]. Both outcomes were summarised descriptively. We also calculated the percentage of participants with excessive sedentary behaviour, defined as >540 min/day [19].

Participants’ physical activity preferences, barriers, and motivating factors were assessed using a modified version of a questionnaire that has been used in people with SMI [20]. Response options were ‘Disagree’, ‘Neutral’, and ‘Agree’. In the preferences section, participants were also asked to indicate their preferred times for physical activity by ticking all relevant response options from the following: ‘Before 8 a.m.’, ‘8–11 a.m.’, ’11 a.m.–2 p.m.’, ‘2–5 p.m.’, ‘5–8 p.m.’, ‘After 8 p.m.’, and ‘No preference’. In the barriers section, nine new items were added after the pre-pandemic wave of recruitment to ensure coverage of all the domains of the Theoretical Domains Framework [21] and COM-B model of behaviour [22]: ‘I don’t have anyone to do physical activity with’, ‘The thought of physical activity makes me worry’, ‘I don’t think that physical activity will benefit me’, ‘My close contacts don’t support or encourage my physical activity’, ‘I get easily distracted from the physical activity I have planned’, ‘I don’t make plans for doing physical activity’, ‘I don’t think physical activity is important’, ‘Physical activity is not something I do automatically’, and ‘Also fear of judgement/stigma because of mental (or physical) health problems’ [23,24].

The health section of the survey included questions on self-rated mental health in the last 12 months (‘Excellent’, ‘Good’, ‘Moderate’, ‘Poor’, or ‘Very poor’), smoking behaviour, consumption of fruit and vegetables, and height and weight (from which body mass index was calculated). Fatigue was assessed using the following two questions: ‘What was your average fatigue level in the last 2 weeks?’ (5-point ordinal scale ranging from 0 [No fatigue] to 4 [Extreme fatigue]) and ‘How much of your waking time have you felt fatigued in the last 2 weeks?’ (5-point ordinal scale ranging from 0 [None of the time] to 4 [All of the time]). A composite fatigue score was calculated by multiplying the responses to these two questions. Depression severity was assessed using the Patient Health Questionnaire-8 [25].

### 2.6. Data Analysis

Survey data were organised and cleaned in Microsoft Excel. Analyses were conducted using Stata 17 software (StataCorp, 2021. Stata Statistical Software: Release 17. College Station, TX, USA: StataCorp LLC.).

Descriptive statistics (e.g., % [*n*], mean [SD], median [IQR]) were used to summarise participant characteristics and responses to the physical activity and sedentary behaviour questions and answer the following research questions: 1. What are the patterns of physical activity and sedentary behaviour in people with SMI?; 3. What are the physical activity preferences of people with SMI?; 4. What barriers to participating in physical activity are reported by people with SMI?; 5. What motivating factors for participating in physical activity are reported by people with SMI?

To answer research question 6 (Do physical activity preferences, barriers and motivating factors differ by gender, age group, physical activity status, and mental health status in this population?), we highlighted factors where there was a ≥10% absolute difference in the proportions between the groups.

Univariable and multivariable logistic regression analyses were performed to identify factors associated with being sufficiently active (dependent variable) and answer research question 2: What are the associations between physical activity status and selected sociodemographic and health variables in this population? Being sufficiently active was defined in three ways [26]:≥150 min/week aerobic physical activity of at least a moderate intensity, including walking (not sufficiently active: <150 min/week)≥150 min/week aerobic physical activity of at least a moderate intensity, excluding walking (not sufficiently active: <150 min/week)≥2 sessions/week muscle-strengthening activity (not sufficiently active: <2 sessions/week)

Independent variables included age, gender, ethnicity, employment status, body mass index, smoking status, mental health status, and fatigue. Selection of these variables was based on a literature review. Multivariable logistic regression included that factor of interest and adjusted for covariates of gender, age category, ethnicity, employment category, body mass index, fatigue category, smoking status, and mental health status. Odds ratios and 95% CIs are reported.

All analyses were performed using available data, and differences in characteristics between participants with and without completed physical activity were compared. All statistical analyses described above were pre-specified and uploaded on the Open Science Framework (https://osf.io/rgc62/; accessed on 29 July 2022).

## 3. Results

There were 529 survey respondents. The majority (71%) of respondents completed the second version of the survey that had the nine additional barriers items. The mode of completion was as follows: post, *n* = 196; online, *n* = 106; phone or in-person, *n* = 186; unknown, *n* = 41. Those who completed the survey via post or online did so independently.

Table 1 presents a summary of participant characteristics. The mean age was 49.3 years (SD 13.1 years, range 20–86; *n* = 519), 58% were male, and the majority were ‘not professionally active’ (67%) and white (79%). The mean body mass index was 31.2 kg/m^2^ (SD 7.8 kg/m^2^, range 15–68, *n* = 437). Two-thirds (67%) of participants reported having slight to moderate fatigue, and 30% reported having poor-to-very poor mental health.

### 3.1. What Are the Patterns of Physical Activity and Sedentary Behaviour in People with SMI? (RQ1)

Table 2 presents a summary of the physical activity and sedentary behaviour data. There was 76% of participants classified as active (≥150 min/week of moderate-to-vigorous physical activity) when walking time was included, decreasing to 57% when walking time was excluded. There was a finding of 28% of participants who reported engaging in ≥2 sessions per week of muscle strengthening activity. Mean sedentary time was 387.5 min/day (SD 253.6 min/day), with 26% of respondents reporting excessive sedentary behaviour (≥540 min/day).

### 3.2. What Are the Associations between Physical Activity Status (Defined as Sufficiently Active or Not) and Selected Sociodemographic and Health Variables in this Population? (RQ2)

Table 3 shows the prevalence of sufficiently active status by selected participant characteristics.

#### 3.2.1. Unadjusted Associations

Variables that were significantly associated with aerobic physical activity status included employment status, body mass index, smoking status, fatigue, mental health, and depression. A greater percentage of sufficiently active status was observed for:Professionally active vs. not professionally active (inc. walking): 90% vs. 76%Professionally active vs. not professionally active (exc. walking): 74% vs. 57%Healthy and underweight (BMI <25 kg/m^2^) vs. with obesity (BMI ≥30 kg/m^2^) (inc. walking): 85% vs. 73%Never smoked vs. previous smoker and current smoker (inc. walking): 85% vs. 76% vs. 74%No fatigue vs. severe fatigue (inc. walking): 86% vs. 67%Excellent-good mental health vs. poor-very poor mental health (inc. walking): 84% vs. 72%Excellent-good mental health vs. poor-very poor mental health (exc. walking): 65% vs. 52%Minimal depression vs. severe depression (inc. walking): 85% vs. 57%Minimal depression vs. severe depression (exc. walking): 63% vs. 40%

Variables that were significantly associated with muscle strengthening activity status included gender, age, mental health, and depression. A greater percentage of sufficiently active status was observed for:Male vs. female: 33% vs. 22%Age 18–34 years vs. ≥65 years: 39% vs. 21%Excellent-good mental health vs. moderate mental health and poor-very poor mental health: 39% vs. 28% vs. 18%Minimal depression vs. moderate depression and severe depression: 35% vs. 20% vs. 22%

#### 3.2.2. Adjusted Associations

Variables that were found to be independently associated with aerobic physical activity status included employment status and consumption of fruit and vegetables. Professionally inactive participants were less likely to perform sufficient aerobic physical activity than professionally active participants (inc. walking: OR = 0.39, 95% CI 0.18–0.89; exc. walking: OR = 0.49, 95% CI 0.27–0.88). People who consumed ≥5 portions of fruit and vegetables per day were more likely to be sufficiently active than those who consumed fewer portions (exc. walking: OR 1.92, 95% CI 1.02–3.62).

Variables that were found to be independently associated with muscle strengthening activity status included age and mental health. Participants aged ≥65 years were less likely to do sufficient muscle strengthening activity than participants aged 18–34 years (OR = 0.27, 95% CI 0.11–0.69), as were people who reported poor-very poor mental health versus excellent-good mental health (OR 0.35, 95% CI 0.18–0.65).

### 3.3. What Are the Physical Activity Preferences of People with SMI and Do They Differ by Gender, Age Group, Physical Activity Status, and Mental Health Status? (RQ3, RQ6)

Figure 1 and Figure 2 show the physical activity preferences of the group as a whole. More than 60% of the participants stated that they preferred activities that could be accomplished on their own, at their own time and pace, and close to home. The most preferred physical activity type was low intensity activities, such as walking (76%). Approximately one-half of the participants disagreed with the statements about sports and high-intensity and strength-based activities. The most preferred source of support for physical activity was social support from family and friends (54%). The least preferred sources of support were a support worker or psychologist/counsellor. The most preferred time to do physical activity was 11 a.m.–2 p.m. (40%), whereas the least preferred was before 8 a.m. (10%).

Table 4 shows the preference ratings (% agree) by selected participant characteristics. The shaded cells indicate factors where there was a ≥10% absolute difference in the proportions between the groups. There were a few noteworthy sub-group differences. Females were more likely to prefer sports. Older adults and people with poor-very poor mental health were less likely to prefer activities performed in a group. Older adults were less likely to prefer activities that are supervised. Older adults and people who were inactive or had poor-very poor mental health were less likely to prefer activities that are high intensity, strength-based, or sports. For a more detailed account of the preference data, see Appendix A.

### 3.4. What Barriers to Participating in Physical Activity Are Reported by People with SMI and Do They Differ by Gender, Age Group, Physical Activity Status, and Mental Health Status? (RQ4, RQ6)

Figure 3 shows the physical activity barriers data for the group as a whole. The most commonly cited barriers were poor physical and mental health (50% and 54%, respectively), lack of motivation (55%), and being too tired (50%). Those ≥60% disagreed that the following were barriers: not having the right clothes/shoes (61%); fear of injury (60%); the mental health service taking up too much of their time (71%); work demands (68%); lack of access to childcare (77%); thinking that physical activity will not benefit them (65%) or is unimportant (64%).

Table 5 shows the barrier ratings (% agree) by selected participant characteristics. The shaded cells indicate factors where there was a ≥10% absolute difference in the proportions between the groups. There were a few noteworthy sub-group differences. Females were less likely to report their weight as a barrier. Older adults were less likely to report a lack of money, motivation, or time as barriers. Inactive people were more likely to report the following as barriers: poor physical health, lack of motivation, not liking the gym scene, their weight, exercise making them exhausted, and not enjoying physical activity. Higher % agree ratings were observed for people with poor-very poor mental health across most of the barriers. For a more detailed account of the barrier data, see Appendix A.

### 3.5. What Motivating Factors for Participating in Physical Activity Are Reported by People with SMI and Do They Differ by Gender, Age Group, Physical Activity Status, and Mental Health Status? (RQ5, RQ6)

Figure 4 shows the motivating factors data for the group as a whole. Eleven factors had a rate of agreement ≥60%, with the five highest ranking factors being ‘to improve my mental health’ (84%), ‘to get stronger and fitter’ (81%), ‘to get out of the house’ (79%), ‘to have more energy’ (77%), and ‘to cope with life stressors better’ (74%).

Table 6 shows the motivating factor ratings (% agree) by selected participant characteristics. The shaded cells indicate factors where there was a ≥10% absolute difference in the proportions between the groups. There were a few noteworthy sub-group differences. Preventing sickness and illness was a more common motivating factor for females and people with excellent-good mental health. Younger adults were more likely to cite health and fitness benefits as motivating factors. People with poor-very poor mental health were less likely to cite making new friends and spending time with others as motivational factors. For a more detailed account of the motivating factors data, see Appendix A.

## 4. Discussion

Our survey enabled an in-depth exploration of physical activity in people with SMI, covering patterns, preferences, barriers, and motivating factors. It is one of the largest studies of its kind. Key findings are as follows: (i) large proportions of the sample were insufficiently active and excessively sedentary; (ii) respondents were less likely to have sufficient aerobic physical activity if they were professionally inactive or consumed fewer than five portions of fruit and vegetables per day, and less likely to have sufficient muscle strengthening activity if they were aged ≥65 years or had poor mental health; (iii) over 60% of participants preferred physical activity that could be performed on their own, at their own time and pace, and close to home; (iv) low intensity activities such as walking were by far the most preferred type of physical activity; (v) the most commonly endorsed source of support was social support from family and friends, whereas relatively few preferred support from a support worker or psychologist/counsellor; (vi) common motivating factors for physical activity included ‘to improve my mental health’, ‘to get stronger and fitter’, and ‘to have more energy’, however, poor mental health, poor physical health, and being too tired were also common barriers.

### 4.1. Patterns of Physical Activity and Sedentary Behaviour in People with SMI

Our study provides new information on the proportions of adults with SMI who meet public health recommendations for sufficient aerobic and muscle strengthening activity. There was a finding of 43% of participants who did not meet the recommendation of 150 min of moderate-to-vigorous aerobic activity per week. Worldwide, 31.1% of adults are estimated to have insufficient aerobic activity [27], so our findings support the notion that people with SMI are less likely to meet physical activity guidelines than people without SMI [28]. Our data also indicate that these people are more sedentary. For example, 26% of the current sample reported excessive sedentary behaviour (>9 h/day) compared with ~9% for the general adult population from the Health Survey for England data [19]. In the general worldwide population, it is estimated that a decrease of 10% in the number of people not meeting these guidelines could result in averting 533,000 premature deaths each year [7]. Supporting people with SMI to be less sedentary and more physically active should therefore be a global public health priority. Our findings support recent calls to expand individual-focused and community-level interventions at a global level in order to reduce excess mortality in people with SMI [29,30].

To our knowledge, this is the first study to provide data on participation in muscle strengthening activity amongst adults with SMI. The majority of participants (58%) reported no muscle strengthening activity and only 28% met the guideline recommendation of ≥2 sessions per week. That substantially fewer participants met this recommendation compared with the aerobic activity recommendation (28% vs. 57%, respectively) is consistent with data from the general population [31], and may reflect the fact that muscle strengthening activity has historically received less emphasis in public health campaigns [32]. Strong clinical and emerging epidemiological evidence shows that muscle strengthening activity is independently associated with multiple health outcomes, including a reduced risk of all-cause mortality, the incidence of diabetes, and enhanced cardiometabolic, musculoskeletal and mental health [32]. Physical activity initiatives for people with SMI should therefore include efforts to promote regular muscle strengthening activity.

Our exploration of factors associated with regular physical activity participation also highlights three key groups for policy focus and intervention. Firstly, specific attention could be given to those who are professionally inactive given the high prevalence of this factor (67%), its association with insufficient aerobic activity, and the fact that health risk behaviours tend to cluster in people who are unemployed [33]. Secondly, the low prevalence of sufficient muscle strengthening activity in those who were over the age of 65 (21%) is concerning because muscle strength is of particular importance for older adults, for example for maintaining functional independence and a low risk of falling [34,35]. Thirdly, people with poor mental health could also be specifically targeted given that they were less likely to engage in regular physical activity and more likely to cite a range of barriers to participation, alongside evidence that physical activity can improve a range of mental health outcomes [10].

### 4.2. Physical Activity Preferences of People with SMI

Few studies have explored physical activity preferences and associated factors in people with SMI. Chapman et al. [20] assessed preferences in 142 community-dwelling adults with mental illness (SMI and other conditions), exploring associations with psychological distress. Fraser et al. [36] assessed preferences among 101 inpatient adults with mental illness (SMI and other conditions), exploring associations with psychological distress and gender. Romain et al. [37] assessed preferences in a mixed group of inpatients and outpatients with SMI (*n* = 114), exploring associations with age, gender, BMI, and physical activity stage of change. We assessed preferences in 529 adults with SMI living in the community, exploring associations with age, gender, physical activity status, and mental health status. Our findings, therefore, provide additional and nuanced insights. They also build on previous reviews of interventions to promote physical activity in this population [38,39] and identify some important factors for researchers, practitioners, and commissioners to consider when co-designing interventions.

With the exception of social support (e.g., from friends/family; 54%), we observed that none of the potential sources of support for physical activity were widely endorsed by participants. The least support was found for psychologist/counsellor or support worker engagement in physical activity programmes. This adds to previous studies that have suggested engaging people with SMI in physical activity, it is important to avoid replicating elements of the clinical environment as this reinforces any associated stigma [40]. Instead, interventions should seek to focus on creating opportunities for participants to share ‘connection stories’ as this can help build community and resilience [41]. There is also a need to find ways to break down the hierarchical relationship between service deliverers and users, which may help deliver physical and psycho-social benefits [40].

The preference for activities to take place ‘close to home’ (61%) is consistent with previous data [20] but may present challenges for implementation when viewed through a health inequalities lens. People living in socio-economically deprived communities have limited access to physical activity facilities [42], and yet there is almost three times the prevalence of SMI in these communities when compared with more affluent areas [43]. The presence of these spatial inequalities, if not considered and adequately addressed by interventions and policy-makers, risks exacerbating the health inequalities that exist for people with SMI.

The preference for sporting activities reported by a higher proportion of females than males (36% vs. 19%) is surprising and counter to data in the general population [18]. In the UK, there has been a sustained focus on the promotion of sport for women through national campaigns [44], and there is some encouraging evidence that trends towards the engagement of women and girls in sport is changing [45]. The extent to which our findings apply more broadly to women with SMI is unknown but our data are encouraging. To ensure interventions capitalise on this potential change in attitudes, recent consensus regarding the framing of sport and physical activity messaging, especially for marginalised groups, highlights the importance of co-production [46] to ensure inclusivity and avoid further stigma through unintended but nevertheless discriminatory messages [47].

### 4.3. Barriers and Motivating Factors for Physical Activity of People with SMI

Our data on barriers and motivating factors are generally consistent with a previous systematic review and meta-analysis on this topic [11]. Among specific aspects of health and well-being, the most common motivations were ‘to improve my mental health’ (84%), ‘to get stronger and fitter’ (81%), and ‘to have more energy’ (77%). However, poor mental health, poor physical health, and being too tired were also identified as common barriers towards physical activity (all ≥50%). Taking this into account, physical activity programmes for people with SMI should be designed to improve both physical and mental health and well-being, while also providing the necessary levels of supervision or assistance for each individual to overcome any physical and psychological barriers and achieve their goals. Employing activity pacing techniques might also be useful to help individuals to manage their energy levels and avoid over- or under-activity [48]. People with poor-to-very poor mental health may require additional support to manage barriers, since this sub-group was more likely to report most of the barriers. Given the diverse range of physical activity attitudes identified, interventions may need to be designed with the flexibility to address individual needs and preferences.

### 4.4. Implications

Physical activity action plans must be adequately resourced, monitored, and enforced to truly advance the fundamental rights of people with SMI to fully participate in physical activity [49]. These plans should include research to develop and evaluate new physical activity interventions for this population. For example, a recent systematic review found that, among 32 intervention studies, very few had specifically set out with the primary aim to increase physical activity, most had a small sample size, few had measured changes in physical activity using objective measures, and results were mostly equivocal on whether physical activity can be changed in people with SMI [38]. We are currently working to address this need through a programme of research called SPACES (Supporting Physical Activity through Co-production in people with Severe mental ill health), which is funded by the UK National Institute for Health and Care Research [50]. The current survey findings have been used as part of an intervention co-production process, and we are currently preparing to pilot the intervention ahead of conducting a full-scale randomised controlled trial.

### 4.5. Limitations

The inclusion of data on physical activity from a large sample of people with SMI is a major strength of this study. However, the current findings should be interpreted with some caution due to several methodological considerations. First, all the measures of this study were self-reported. Although self-reports have been widely used for gaining insight into the physical activity levels of populations [51], they have limited validity [52] and issues of recall and response bias (e.g., social desirability, inaccurate memory) [53]. Second, despite efforts to ensure that the cohort was as representative as possible of people with SMI in the UK, it may be that there are differences between the cohort participants and the general SMI population. However, the fact that the results are mostly consistent with the existing literature is encouraging. Third, the study was exploratory in nature; we did not include statistical hypothesis testing, and instead present raw proportions, odds ratios, and confidence intervals. There is the potential that observed differences in sub-groups and between factors are chance observations. Fourth, we did not ask about access or proximity to services or physical activity opportunities. Fifth, we did not collect data on an individual’s specific SMI diagnosis.

## 5. Conclusions

The present study investigated physical activity attitudes and behaviours in adults with SMI, living in the community. The findings support the existing literature that people with SMI engage in less physical activity and more sedentary behaviour than the general population. However, the findings also indicate that people with SMI are interested in doing physical activity, with improving mental health, physical fitness, and energy levels being commonly cited motivators. Programmes that include low-intensity activities such as walking, optional social and healthy lifestyle components, and that are led by an instructor who can provide assistance with motivation and overcoming physical and mental health barriers experienced by adults with SMI, may be best able to accommodate the physical activity preferences of this group. People with poor mental health may require more support with overcoming barriers. Given the diverse range of physical activity attitudes identified, interventions may need to be designed with flexibility for individual needs and preferences and the incorporation of a range of behaviour change techniques. The information from this study has the potential to guide physical activity intervention planning and programming, which could contribute to decreasing the chronic disease burden, and improving psychological well-being in adults with SMI.

## Figures and Tables

**Figure 1 ijerph-20-02548-f001:**
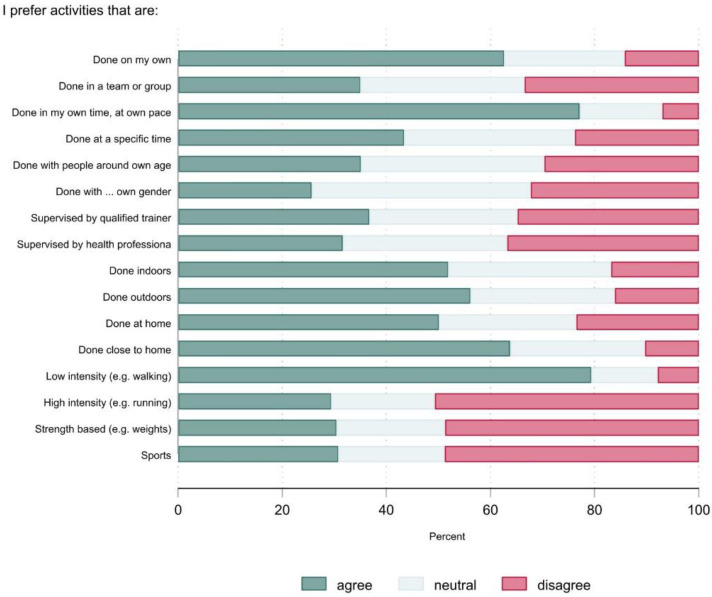
Physical activity preferences (*n* = 529).

**Figure 2 ijerph-20-02548-f002:**
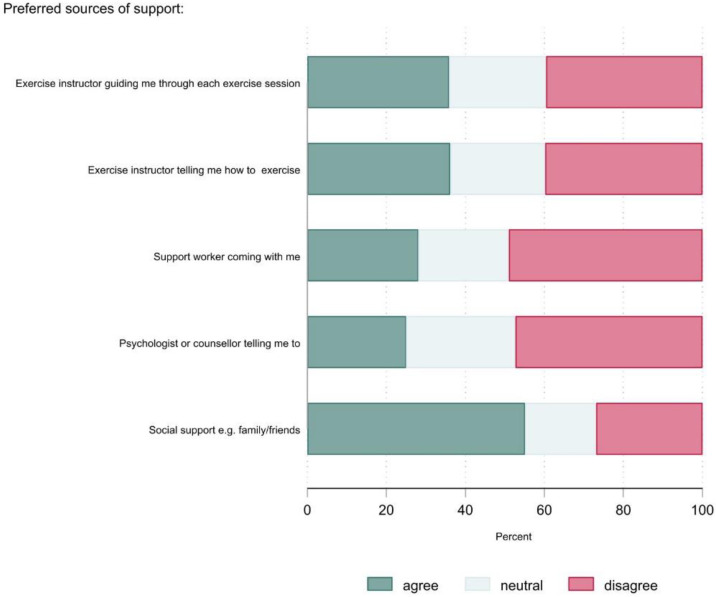
Preferred sources of support (*n* = 529). Social support was asked in the second version of the survey only.

**Figure 3 ijerph-20-02548-f003:**
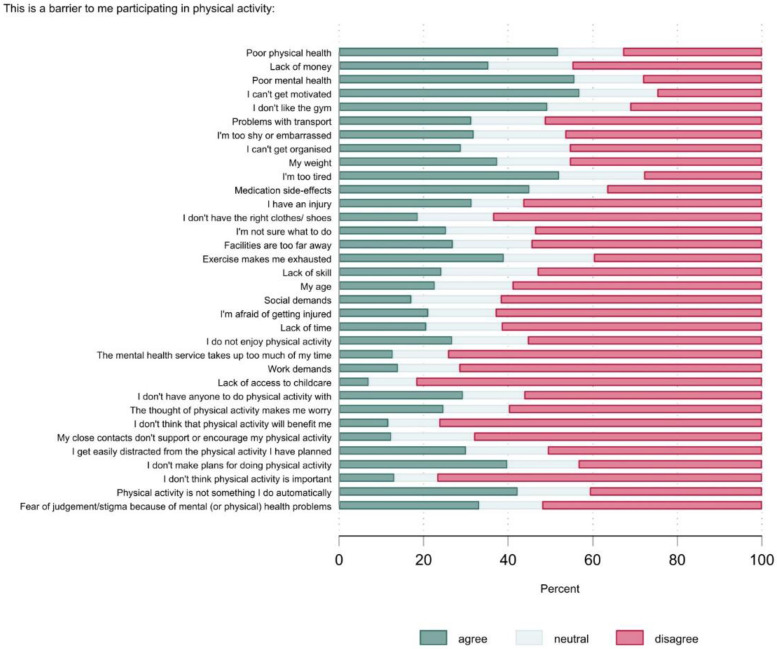
Physical activity barriers (*n* = 529).

**Figure 4 ijerph-20-02548-f004:**
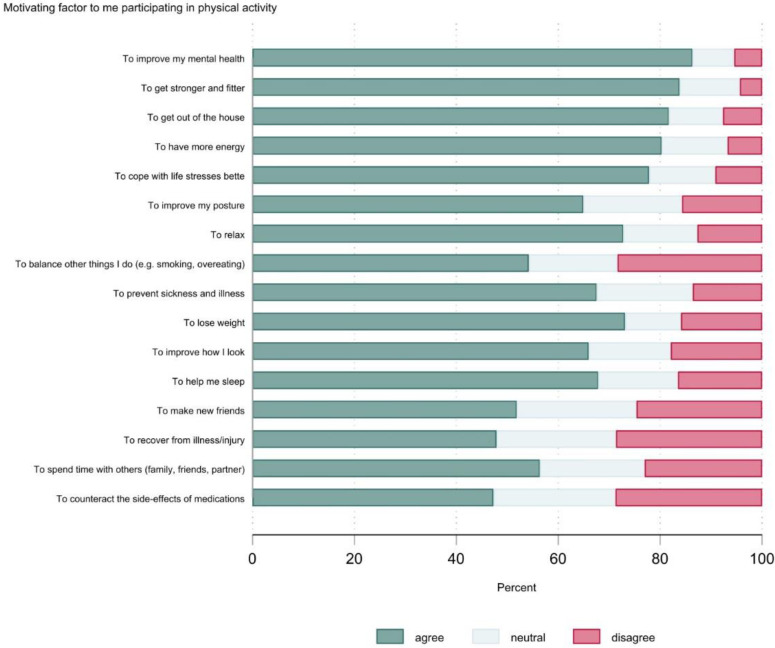
Motivating factors for physical activity (*n* = 529).

**Table 1 ijerph-20-02548-t001:** Participant characteristics (*n* = 529).

Characteristics		% (*n*)
Age	18–34 years	15 (81)
	35–64 years	70 (371)
	>65 years	13 (67)
	Missing	2 (10)
Gender	Male	58 (308)
	Female	40 (212)
	Transgender	1 (3)
	Preferred not to say	1 (4)
	Missing	0 (2)
Ethnicity	White—UK/Irish/other	79 (416)
	Asian	4 (21)
	Black/African/Caribbean	5 (26)
	Mixed multiple groups	3 (14)
	Other	2 (12)
	Missing	8 (40)
Employment	Professionally active	19 (99)
	Not professionally active	67 (356)
	Other	4 (21)
	Missing	10 (53)
Body mass index	<18.5 kg/m^2^	0 (2)
	18.5–24.9 kg/m^2^	18 (93)
	25–29.9 kg/m^2^	22 (119)
	≥30 kg/m^2^	42 (223)
	Missing	17 (92)
Self-rated mental health	Excellent-good	38 (199)
	Moderate	31 (166)
	Poor-very poor	30 (157)
	Missing	1 (7)
PHQ-8 depression severity	0–4 (minimal)	30 (159)
	5–9 (mild)	23 (120)
	10–14 (moderate)	16 (85)
	15–19 (moderately severe)	14 (72)
	20–24 (severe)	9 (47)
	Missing	9 (46)
Smoking	Smoker	35 (185)
	Used to smoke	31 (166)
	Never smoked	32 (171)
	Missing	1 (7)
Consumption of fruit and vegetables	≥5 portions per day	17 (91)
	≤4 portions per day	82 (433)
	Missing	1 (5)
Fatigue	0 (none)	13 (71)
	1–10 (slight to moderate)	68 (359)
	11–20 (severe)	14 (73)
	Missing	5 (26)

PHQ-8, Patient Health Questionnaire-8. Percentages have been rounded to the nearest whole number.

**Table 2 ijerph-20-02548-t002:** Descriptive statistics for physical activity and sedentary behaviour (*n* = 529).

Variable	% (*n*)	Minutes per Day		Minutes per Week	
		Mean (SD)	Median (IQR)	Mean (SD)	Median (IQR)
Time spent in vigorous PA	97 (515)	12 (25)	0 (0, 21)	85 (176)	0 (0, 105)
Time spent in moderate PA, inc. walking	95 (501)	58 (68)	39 (17, 72)	403 (477)	270 (120, 510)
Time spent in moderate PA, exc. walking	96 (510)	28 (44)	15 (4, 34)	197 (306)	105 (30, 240)
Time spent walking	97 (515)	31 (52)	18 (9, 39)	216 (361)	125 (60, 270)
Time spent in SB	90 (478)	388 (254)	360 (180, 600)		
Time spent in bed per night	91 (483)	522 (138)	510 (450, 600)		
Active ≥150 min/week, including walking	76 (400)				
Active ≥150 min/week, excluding walking	57 (304)				
Active <30 min/week, including walking	5 (29)				
Active <30 min/week, excluding walking	18 (96)				
Sedentary ≥540 min/day	26 (136)				
Muscle strengthening sessions					
0 per week	58 (309)				
1 per week	10 (52)				
≥2 per week	28 (147)				
Missing	4 (21)				

PA, physical activity; SB, sedentary behaviour.

**Table 3 ijerph-20-02548-t003:** Prevalence of sufficiently active status by participant characteristic.

Variable	Active ≥ 150 min/week inc. Walking ^a^			Active ≥ 150 min/week exc. Walking ^b^			MSA ≥ 2 sessions/week ^c^		
	% (*n*)	Unadjusted ^d^ OR (95% CI)	Adjusted ^e^ OR (95% CI)	% (*n*)	Unadjusted ^d^ OR (95% CI)	Adjusted ^e^ OR (95% CI)	% (*n*)	Unadjusted ^d^ OR (95% CI)	Adjusted ^e^ OR (95% CI)
Gender									
Female	78 (159/203)			57 (115/203)			22 (45/203)		
Male	79 (237/301)	1.02 (0.66–1.58)	1.22 (0.70–2.13)	62 (186/300)	1.25 (0.87–1.79)	1.41 (0.88–2.25)	33 (97/297)	1.70 (1.13–2.57)	1.33 (0.79–2.23)
Ethnicity									
White	78 (312/401)			60 (241/402)			29 (117/400)		
Other than white	83 (60/72)	1.43 (0.73–2.77)	1.04 (0.46–2.33)	61 (43/71)	1.03 (0.61–1.72)	0.87 (0.45–1.69)	31 (22/72)	1.06 (0.62–1.84)	0.91 (0.45–1.83)
Consumption of fruit and vegetables									
≤4 portions per day	77 (323/420)			58 (243/421)			29 (119/416)		
≥5 portions per day	85 (74/87)	1.71 (0.91–3.21)	2.24 (0.97–5.18)	69 (59/86)	1.60 (0.98–2.63)	1.92 (1.02–3.62)	30 (27/89)	1.09 (0.66–1.79)	1.19 (0.64–2.23)
Age (years)									
18–34	86 (70/81)			65 (53/81)			39 (31/80)		
35–64	77 (275/358)	0.52 (0.26–1.03)	0.73 (0.32–1.68)	59 (208/355)	0.75 (0.45–1.24)	0.76 (0.40–1.44)	28 (99/354)	0.61 (0.37–1.02)	0.54 (0.29–1.00)
≥65	73 (47/64)	0.43 (0.19–1.01)	0.55 (0.19–1.56)	55 (36/65)	0.66 (0.34–1.28)	0.98 (0.41–2.32)	21 (14/66)	0.43 (0.20–0.89)	0.27 (0.11–0.69)
Employment									
Professionally active	90 (86/96)			74 (72/97)			37 (35/95)		
Not professionally active	76 (261/345)	0.36 (0.18–0.73)	0.39 (0.18–0.89)	57 (196/345)	0.46 (0.28–0.75)	0.49 (0.27–0.88)	29 (99/346)	0.69 (0.43–1.11)	1.08 (0.60–1.92)
Other	80 (16/20)	0.47 (0.13–0.67)	0.65 (0.12–3.63)	55 (11/20)	0.42 (0.16–1.14)	0.34 (0.10–1.14)	11 (2/19)	0.20 (0.04–0.93)	0.31 (0.06–1.61)
BMI (kg/m^2^)									
<25	85 (77/91)			66 (61/93)			33 (30/91)		
25–29.9	83 (95/115)	0.86 (0.41–1.82)	0.96 (0.41–2.23)	64 (73/114)	0.93 (0.53–1.66)	0.97 (0.51–1.87)	36 (43/118)	1.17 (0.66–2.07)	1.21 (0.62–2.36)
≥30	73 (158/217)	0.49 (0.26–0.93)	0.51 (0.25–1.02)	56 (122/216)	0.68 (0.41–1.13)	0.71 (0.40–1.25)	26 (55/213)	0.71 (0.41–1.21)	0.66 (0.36–1.21)
Smoking									
Never smoked	85 (140/165)			63 (104/164)			33 (53/161)		
Previous smoker	76 (123/162)	0.56 (0.32–0.98)	0.70 (0.36–1.37)	57 (93/163)	0.77 (0.49–1.19)	0.69 (0.40–1.19)	26 (42/160)	0.73 (0.45–1.17)	0.85 (0.48–1.52)
Current smoker	74 (132/179)	0.50 (0.29–0.86)	0.52 (0.26–1.03)	58 (103/177)	0.80 (0.52–1.24)	0.82 (0.47–1.43)	27 (49/181)	0.76 (0.48–1.20)	0.63 (0.35–1.15)
Fatigue score									
0 (none)	86 (60/70)			58 (40/69)			30 (20/67)		
1–10 (slight to moderate)	79 (277/350)	0.63 (0.31–1.30)	0.84 (0.35–2.01)	61 (215/350)	1.15 (0.68–1.95)	1.53 (0.78–2.99)	30 (104/346)	1.01 (0.57–1.79)	1.50 (0.73–3.10)
11–20 (severe)	67 (45/67)	0.34 (0.15–0.79)	0.56 (0.19–1.63)	49 (33/67)	0.70 (0.36–1.38)	1.62 (0.66–3.96)	22 (16/72)	0.67 (0.31–1.44)	1.17 (0.45–3.04)
Self-rated mental health									
Excellent–good	84 (162/194)			65 (125/191)			39 (74/191)		
Moderate	78 (126/162)	0.69 (0.41–1.17)	0.77 (0.40–1.50)	60 (98/162)	0.81 (0.52–1.25)	0.61 (0.35–1.06)	28 (45/160)	0.62 (0.39–0.97)	0.60 (0.34–1.05)
Poor–very poor	72 (107/149)	0.50 (0.30–0.85)	0.63 (0.32–1.24)	52 (78/151)	0.56 (0.36–0.87)	0.59 (0.33–1.05)	18 (27/151)	0.34 (0.21–0.57)	0.35 (0.18–0.65)
PHQ-8 depression score									
0–4 (minimal)	85 (133/156)			63 (99/156)			35 (55/155)		
5–9 (mild)	76 (90/118)	0.56 (0.30–1.03)	0.73 (0.33–1.65)	64 (75/118)	1.00 (0.61–1.65)	1.39 (0.72–2.70)	30 (35/117)	0.78 (0.46–1.30)	0.74 (0.37–1.48)
10–14 (moderate)	75 (60/80)	0.52 (0.26–1.02)	0.60 (0.24–1.51)	56 (45/81)	0.72 (0.42–1.24)	0.87 (0.40–1.87)	20 (16/82)	0.44 (0.23–0.83)	0.42 (0.18–1.03)
15–19 (moderate/severe)	80 (55/69)	0.68 (0.33–1.42)	0.79 (0.29–2.15)	59 (40/68)	0.82 (0.46–1.47)	1.13 (0.49–2.60)	26 (18/70)	0.63 (0.34–1.18)	0.61 (0.25–1.48)
20–24 (severe)	57 (25/44)	0.23 (0.11–0.48)	0.36 (0.13–1.06)	40 (18/45)	0.38 (0.19–0.76)	0.66 (0.25–1.72)	22 (10/46)	0.44 (0.23–0.83)	0.42 (0.18–1.03)

BMI, body mass index; CI, confidence interval; MSA, muscle strengthening activity; OR, odds ratio; PHQ-8, Patient Health Questionnaire-8. ^a^ reference group; active <150 min/week including walking, ^b^ reference group: active <150 min/week excluding walking, ^c^ reference group <2 MSA sessions/week, ^d^ calculated from univariable logistic regression, ^e^ calculated from multivariable logistic regression adjusting for gender, age category, ethnicity, employment category, body mass index, fatigue category, smoking status, and mental health status (excluding the characteristic being investigated).

**Table 4 ijerph-20-02548-t004:** Physical activity preferences by selected participant characteristics.

		Gender		Age			Physical Activity Status		Mental Health Status		
	Total	Male	Female	18–34	35–64	≥65	Active *	Inactive	Excellent–Good	Moderate	Poor–Very Poor
% (*n*)	(*n* = 529)	(*n* = 212)	(*n* = 308)	(*n* = 81)	(*n* = 371)	(*n* = 67)	(*n* = 304)	(*n* = 206)	(*n* = 199)	(*n* = 166)	(*n* = 157)
I prefer activities that are …											
Done on my own	60 (318)	54 (114)	66 (202)	65 (53)	59 (218)	60 (40)	62 (189)	59 (122)	56 (111)	66 (109)	60 (94)
Done in a team or group	33 (177)	31 (66)	36 (110)	36 (29)	35 (129)	22 (15)	37 (112)	29 (60)	45 (89)	34 (57)	20 (31)
Done in my own time, at my own pace	74 (391)	75 (159)	74 (227)	75 (61)	74 (276)	70 (47)	72 (219)	77 (159)	70 (140)	76 (126)	76 (119)
Done at a specific time	41 (219)	39 (82)	44 (135)	48 (39)	43 (160)	25 (17)	44 (134)	39 (81)	40 (80)	45 (74)	39 (62)
Done with people around my own age	34 (179)	28 (59)	38 (118)	37 (30)	31 (116)	39 (26)	38 (115)	30 (61)	41 (82)	36 (59)	24 (37)
Done with people of my own gender	24 (129)	25 (53)	23 (72)	35 (28)	23 (84)	18 (12)	25 (77)	24 (49)	24 (48)	24 (40)	25 (39)
Supervised by a qualified trainer	35 (185)	35 (74)	36 (110)	44 (36)	35 (129)	22 (15)	32 (98)	39 (81)	41 (82)	31 (52)	31 (48)
Supervised by a health professional	30 (159)	30 (63)	31 (96)	28 (23)	32 (119)	18 (12)	24 (74)	37 (77)	32 (64)	31 (52)	27 (42)
Done indoors	50 (263)	53 (112)	48 (148)	52 (42)	51 (191)	37 (25)	49 (150)	51 (105)	49 (97)	54 (90)	46 (72)
Done outdoors	54 (287)	50 (106)	58 (178)	58 (47)	52 (194)	63 (42)	55 (167)	54 (112)	56 (112)	60 (99)	45 (71)
Done at home	47 (250)	49 (103)	46 (142)	48 (39)	48 (177)	39 (26)	47 (143)	48 (98)	47 (93)	51 (84)	45 (71)
Done close to home	61 (322)	64 (135)	59 (181)	63 (51)	61 (227)	57 (38)	61 (186)	63 (129)	58 (116)	67 (112)	57 (90)
Low intensity (e.g., walking)	76 (404)	79 (167)	76 (233)	69 (56)	78 (288)	82 (55)	76 (232)	79 (162)	78 (155)	78 (130)	74 (116)
High intensity (e.g., running, aerobics)	28 (149)	27 (58)	29 (89)	56 (45)	24 (90)	16 (11)	38 (116)	15 (30)	34 (67)	28 (46)	23 (36)
Strength based (e.g., weight training)	29 (153)	25 (53)	32 (98)	51 (41)	27 (102)	10 (7)	36 (110)	19 (40)	36 (72)	27 (45)	22 (35)
Sports	29 (152)	19 (41)	36 (110)	53 (43)	27 (99)	12 (8)	36 (109)	20 (41)	39 (78)	27 (44)	18 (28)
Preferred sources of support:											
Exercise instructor guiding me through each session	34 (181)	39 (82)	31 (97)	35 (28)	35 (130)	25 (17)	32 (97)	38 (79)	40 (80)	31 (51)	31 (48)
Exercise instructor telling me how to exercise	35 (183)	37 (79)	33 (102)	40 (32)	34 (127)	27 (18)	33 (101)	37 (77)	42 (83)	34 (57)	27 (43)
Support worker coming with me	27 (142)	23 (49)	29 (90)	27 (22)	28 (105)	18 (12)	23 (70)	33 (69)	27 (54)	27 (44)	28 (44)
Psychologist or counsellor telling me to	24 (126)	18 (38)	29 (88)	27 (22)	25 (94)	12 (8)	22 (68)	25 (52)	21 (42)	23 (39)	29 (45)
Social support (e.g., family/friends) ^a^	54 (201)	53 (68)	55 (132)	71 (51)	51 (130)	43 (16)	55 (118)	53 (78)	59 (84)	50 (58)	52 (59)
I prefer doing activity ^b^:											
Before 8 am	10 (55)	9 (20)	11 (33)	9 (7)	11 (40)	9 (6)	13 (38)	7 (14)	13 (26)	10 (17)	8 (12)
8–11 am	32 (170)	36 (77)	30 (91)	28 (23)	32 (117)	40 (27)	34 (104)	29 (59)	29 (58)	37 (61)	31 (49)
11 a.m.–2 p.m.	40 (209)	42 (90)	38 (118)	35 (28)	40 (149)	40 (27)	41 (125)	37 (76)	41 (81)	39 (65)	37 (58)
2–5 p.m.	34 (182)	33 (69)	35 (109)	30 (24)	35 (129)	40 (27)	34 (104)	34 (70)	33 (65)	36 (60)	34 (53)
5–8 p.m.	21 (111)	21 (45)	20 (63)	30 (24)	22 (80)	9 (6)	22 (68)	18 (38)	20 (40)	18 (30)	25 (40)
No preference	11 (57)	8 (17)	13 (39)	11 (9)	12 (44)	6 (4)	11 (34)	10 (20)	13 (26)	8 (14)	11 (17)

Data are reported as % (*n*) agreement with the statement. Shaded cells indicate factors where there was a ≥10% absolute difference in the proportions between the groups. * Active is defined as ≥150 min/week of aerobic physical activity of at least a moderate intensity (excluding walking). ^a^ asked in second version of survey only, ^b^ can select more than one.

**Table 5 ijerph-20-02548-t005:** Physical activity barriers by selected participant characteristics.

		Gender		Age			Physical Activity Status		Mental Health Status		
	Total	Male	Female	18–34	35–64	≥65	Active *	Inactive	Excellent–Good	Moderate	Poor–Very Poor
% (*n*)	(*n* = 529)	(*n* = 212)	(*n* = 308)	(*n* = 81)	(*n* = 371)	(*n* = 67)	(*n* = 304)	(*n* = 206)	(*n* = 199)	(*n* = 166)	(*n* = 157)
Poor physical health	50 (266)	53 (112)	48 (148)	44 (36)	51 (191)	49 (33)	41 (124)	65 (133)	37 (74)	55 (91)	63 (99)
Lack of money	34 (180)	40 (84)	30 (93)	36 (29)	36 (135)	18 (12)	33 (100)	35 (72)	28 (56)	37 (61)	39 (61)
Poor mental health	54 (285)	56 (119)	53 (162)	52 (42)	57 (212)	37 (25)	51 (156)	59 (121)	29 (58)	59 (98)	80 (125)
I can’t get motivated	55 (290)	58 (123)	53 (164)	62 (50)	57 (211)	37 (25)	48 (145)	67 (138)	40 (80)	57 (94)	71 (111)
I don’t like the gym ‘scene’	48 (252)	53 (112)	44 (136)	40 (32)	49 (183)	46 (31)	41 (126)	56 (116)	38 (76)	49 (82)	59 (92)
Problems with transport	30 (159)	34 (72)	27 (84)	25 (20)	32 (120)	21 (14)	28 (84)	33 (68)	24 (47)	30 (50)	39 (62)
I’m too shy or embarrassed	31 (162)	35 (75)	27 (82)	32 (26)	32 (118)	21 (14)	29 (87)	36 (74)	19 (37)	31 (52)	46 (73)
I can’t get organised	28 (146)	25 (54)	29 (90)	27 (22)	29 (108)	19 (13)	24 (74)	33 (69)	19 (37)	31 (51)	36 (56)
My weight	36 (189)	45 (96)	30 (91)	26 (21)	39 (143)	31 (21)	29 (88)	46 (95)	25 (50)	37 (62)	48 (75)
I’m too tired	50 (266)	55 (116)	48 (147)	51 (41)	52 (193)	42 (28)	45 (138)	58 (119)	36 (72)	50 (83)	69 (108)
Medication side-effects	43 (230)	43 (91)	44 (136)	41 (33)	47 (175)	28 (19)	39 (119)	50 (102)	31 (62)	39 (65)	64 (100)
I have an injury	30 (159)	28 (60)	31 (95)	27 (22)	30 (113)	33 (22)	32 (98)	26 (53)	27 (54)	30 (50)	34 (54)
I don’t have the right clothes/shoes	18 (94)	18 (39)	17 (51)	15 (12)	19 (72)	12 (8)	16 (49)	19 (39)	16 (31)	14 (23)	25 (40)
I’m not sure what to do	24 (128)	25 (53)	23 (72)	22 (18)	26 (98)	15 (10)	20 (60)	31 (64)	20 (40)	22 (36)	33 (52)
Facilities are too far away	26 (136)	30 (64)	22 (69)	21 (17)	27 (99)	24 (16)	24 (72)	30 (61)	23 (46)	21 (35)	34 (54)
Exercise makes me exhausted	37 (198)	37 (78)	38 (117)	32 (26)	39 (144)	37 (25)	27 (83)	52 (107)	28 (56)	39 (64)	49 (77)
Lack of skill	23 (122)	24 (50)	22 (69)	23 (19)	25 (91)	12 (8)	21 (63)	27 (56)	17 (33)	23 (39)	31 (49)
My age	22 (115)	20 (43)	23 (70)	2 (2)	25 (91)	28 (19)	19 (59)	26 (53)	18 (36)	20 (34)	28 (44)
Social demands	16 (86)	18 (39)	15 (46)	17 (14)	18 (65)	7 (5)	16 (48)	18 (37)	13 (26)	14 (24)	22 (35)
I’m afraid of getting injured	20 (106)	17 (37)	22 (67)	12 (10)	22 (81)	18 (12)	16 (49)	26 (53)	18 (36)	17 (29)	26 (41)
Lack of time	20 (104)	23 (48)	18 (55)	31 (25)	18 (68)	13 (9)	18 (56)	23 (47)	19 (37)	22 (37)	19 (30)
I do not enjoy PA	26 (135)	26 (56)	25 (76)	19 (15)	27 (100)	25 (17)	16 (50)	39 (81)	18 (35)	25 (42)	35 (55)
Mental health service takes up too much of my time	12 (64)	8 (18)	15 (46)	10 (8)	14 (52)	3 (2)	11 (33)	14 (28)	13 (26)	8 (14)	15 (24)
Work demands	13 (70)	15 (31)	12 (37)	19 (15)	13 (48)	6 (4)	15 (46)	11 (23)	13 (25)	15 (25)	13 (20)
Lack of access to childcare	7 (35)	8 (16)	6 (19)	9 (7)	6 (24)	3 (2)	7 (20)	7 (14)	6 (11)	5 (9)	10 (15)
Asked in updated version of survey only:											
I don’t have anyone to do PA with	18 (93)	15 (31)	19 (59)	25 (20)	16 (61)	13 (9)	14 (42)	23 (47)	15 (30)	17 (29)	22 (34)
The thought of PA makes me worry	15 (78)	13 (28)	16 (48)	16 (13)	15 (57)	10 (7)	11 (33)	21 (44)	8 (15)	12 (20)	27 (43)
I don’t think that PA will benefit me	7 (37)	5 (11)	8 (26)	7 (6)	8 (29)	1 (1)	5 (15)	11 (22)	7 (14)	6 (10)	8 (13)
My close contacts don’t support or encourage my PA	7 (39)	6 (12)	9 (27)	12 (10)	7 (27)	3 (2)	6 (18)	10 (21)	6 (12)	8 (13)	9 (14)
I get easily distracted from the PA I have planned	18 (95)	17 (37)	19 (57)	27 (22)	17 (64)	12 (8)	15 (46)	22 (46)	12 (23)	20 (34)	24 (37)
I don’t make plans for doing PA	24 (125)	20 (42)	27 (82)	28 (23)	24 (90)	15 (10)	13 (38)	39 (81)	16 (32)	25 (42)	32 (50)
I don’t think PA is important	8 (41)	5 (10)	10 (31)	7 (6)	9 (32)	3 (2)	7 (20)	9 (18)	8 (15)	8 (13)	8 (12)
PA is not something I do automatically	25 (133)	24 (50)	27 (82)	28 (23)	26 (96)	18 (12)	18 (54)	36 (74)	20 (40)	24 (40)	33 (52)
Fear of judgement/stigma because of mental (or physical) health problems	20 (104)	16 (34)	22 (69)	23 (19)	21 (77)	10 (7)	17 (53)	24 (50)	12 (24)	17 (28)	33 (52)

Data are reported as % (*n*) agreement with the statement. Shaded cells indicate factors where there was a ≥10% absolute difference in the proportions between the groups. PA, physical activity. * Active is defined as ≥150 min/week of aerobic physical activity of at least a moderate intensity (excluding walking).

**Table 6 ijerph-20-02548-t006:** Motivating factors for physical activity by selected participant characteristics.

		Gender		Age			Physical Activity Status		Mental Health Status		
	Total	Male	Female	18–34	35–64	>65	Active *	Inactive	Excellent–Good	Moderate	Poor–Very Poor
% (*n*)	(*n* = 529)	(*n* = 212)	(*n* = 308)	(*n* = 81)	(*n* = 371)	(*n* = 67)	(*n* = 273)	(*n* = 229)	(*n* = 199)	(*n* = 166)	(*n* = 157)
To improve my mental health	84 (443)	84 (179)	83 (257)	88 (71)	84 (313)	76 (51)	88 (241)	79 (180)	82 (164)	84 (140)	85 (133)
To get stronger and fitter	81 (431)	81 (171)	82 (252)	88 (71)	81 (300)	78 (52)	86 (236)	77 (176)	84 (168)	78 (130)	82 (128)
To get out of the house	79 (420)	79 (168)	81 (250)	79 (64)	80 (295)	81 (54)	83 (227)	76 (175)	85 (169)	80 (132)	73 (115)
To have more energy	77 (408)	78 (165)	77 (238)	86 (70)	77 (287)	67 (45)	81 (220)	73 (167)	78 (155)	77 (128)	77 (121)
To cope with life stresses better	74 (394)	76 (161)	75 (230)	79 (64)	76 (281)	66 (44)	81 (221)	67 (154)	79 (157)	75 (124)	69 (109)
To improve my posture	62 (328)	60 (127)	64 (196)	65 (53)	63 (232)	57 (38)	64 (176)	61 (139)	68 (135)	59 (98)	58 (91)
To relax	70 (369)	68 (145)	72 (221)	67 (54)	72 (268)	63 (42)	73 (200)	66 (152)	77 (153)	69 (115)	61 (96)
To balance other things I do (e.g., smoking)	52 (274)	52 (110)	52 (159)	58 (47)	54 (199)	33 (22)	55 (151)	47 (108)	53 (105)	50 (83)	54 (84)
To prevent sickness and illness	65 (344)	58 (123)	70 (215)	65 (53)	65 (241)	66 (44)	69 (188)	62 (143)	71 (142)	63 (105)	60 (94)
To lose weight	71 (373)	73 (155)	68 (210)	81 (66)	71 (264)	55 (37)	72 (196)	71 (163)	75 (150)	69 (115)	67 (105)
To improve how I look	64 (336)	62 (132)	64 (198)	81 (66)	63 (233)	45 (30)	64 (176)	65 (148)	70 (140)	58 (97)	61 (95)
To help me sleep	65 (342)	62 (131)	68 (208)	78 (63)	64 (237)	55 (37)	70 (190)	59 (134)	69 (137)	61 (102)	64 (100)
To make new friends	50 (264)	48 (101)	52 (160)	53 (43)	49 (181)	55 (37)	55 (150)	44 (101)	61 (122)	46 (76)	41 (64)
To recover from illness/injury	46 (243)	41 (87)	49 (152)	58 (47)	44 (165)	43 (29)	48 (130)	44 (100)	49 (98)	43 (71)	45 (70)
To spend time with others (family, friends, partner)	54 (285)	53 (113)	56 (171)	60 (49)	53 (197)	54 (36)	57 (155)	50 (114)	62 (123)	49 (82)	49 (77)
To counteract the side-effects of medications	45 (239)	42 (89)	47 (146)	52 (42)	45 (168)	37 (25)	45 (123)	44 (101)	47 (94)	43 (71)	46 (72)

Data are reported as % (*n*) agreement with the statement. Shaded cells indicate factors where there was a ≥10% absolute difference in the proportions between the groups. * ≥150 min/week of aerobic physical activity of at least a moderate intensity (excluding walking).

## Data Availability

The anonymized dataset may be made available on reasonable request. Requests for data may be sent to the Closing the Gap Network email: ctg-network@york.ac.uk whose Steering Committee manages our data access requests.

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
