# Peer review of "Physical Activity in Adults with Schizophrenia and Bipolar Disorder: A Large Cross-Sectional Survey Exploring Patterns, Preferences, Barriers, and Motivating Factors"

_ijerph, 2023, doi:10.3390/ijerph20032548_

Round 1
Reviewer 1 Report
This is a long-awaited study for intervention researchers to make sound decisions on developing new programs to benefit adults with SMI. The literature review is sound, the methods showed creativity around the pandemic era with multiple ways of collecting the survey data. Results and discussion are written with suggestions that do not go beyond the data. Limitations are generous and describe appropriate biases. A well-written, soundly designed study that needs to be shared broadly.
Author Response
This is a long-awaited study for intervention researchers to make sound decisions on developing new programs to benefit adults with SMI. The literature review is sound, the methods showed creativity around the pandemic era with multiple ways of collecting the survey data. Results and discussion are written with suggestions that do not go beyond the data. Limitations are generous and describe appropriate biases. A well-written, soundly designed study that needs to be shared broadly.
Thank you for reviewing our paper and for your positive feedback.
Reviewer 2 Report
My comments are in attachment.

Author Response
The manuscript reports on the investigation of the cross-sectional associations of physical activity patterns, preferences, barriers, and motivating factors with physical activity in persons with severe mental illness. A strength of the study is the large and representative sample. My comments, those of substance and those of an editorial nature are below.
Thank you for reviewing our manuscript.
line 2 – “Mental ill health” should be replaced by mental illness.
There is no consensus about the term that should be used to describe mental health and we prefer to use the term ‘mental ill health’ in line with some of our other publications in this area and after consultation with service users about the term to use.
line 2 – Please add “sedentary behaviour” in the title because authors mentioned in the introduction and results.
Although we provide some data on sedentary behaviour, most of the data pertains to physical activity. As such, we would prefer to leave the title unchanged.
line 26 – The results of the abstract should be modified when authors have to check the data analyses. I would expect the authors to analyse the relationships of walking and sedentary behaviour with physical activity preference, social support, barriers, and motivating factors.
The analyses that we performed address our research questions and were pre-specified in a statistical analysis plan. The categories for physical activity status were based on national guidelines for physical activity, which don’t include thresholds for categorising walking or sedentary behaviour.
lines 67-78 – It is not entirely clear which variables were the independent or dependent variables in the aim of the study. For example, sedentary behaviour should be one of dependent variables, while mental health status should be an independent variable. A rationale is needed. The research questions should be shortened because some words/phrases might be repeated.
Variables are labelled as independent or dependent in the methods. We did not seek to explore the relationship between sedentary behaviour and mental health status in this study. Rather the focus was to inform intervention development by exploring PA patterns, preferences, barriers and motivations. Our preference is to keep the research questions written out in full for clarity.
line 88 – Please provide more information about the characteristics of participants with severe mental illness, i.e., number of participants in each diagnosis, years of documented diagnosis, and years of treatment. How to select participants in the study? How many people are excluded from the study due to incomplete data?
To take part in the study, participants had to have a documented diagnosis of schizophrenia, schizoaffective disorder or bipolar disorder. However, as explained in the limitations section, we did not collect data on individual’s specific SMI diagnoses. The recruitment procedure is explained in section 2.4. All 529 respondents were included. The rates of missing data for specific survey items are shown in the results tables.
line 109 – Relative to the surveys, it is recommended that the following be clarified/addressed:
* Was there a timeframe (e.g., previous month, previous 3 months, and previous years) that participants should base their responses on? (Example: Over the past 3 months, how many times did you engage in moderate and vigorous activities?)
The questions ask for “in an average week”, as described in the methods and shown in the survey which we included a link to.
* Has the validity and reliability of the surveys been established, including the sufficient internal consistency in the separate domains, i.e., sedentary behaviour, physical activity patterns, preferences, barriers, and motivating factors.
The survey included a range of tools and questions that have been used in studies previously. All components are described in the methods with supporting references. For physical activity and sedentary behaviour, references 15 and 17 provide data on validity and reliability:
- Orrell, A.; Doherty, P.; Miles, J.; Lewin, R. Development and Validation of a Very Brief Questionnaire Measure of Physical Activity in Adults with Coronary Heart Disease. European Journal of Cardiovascular Prevention & Rehabilitation 2007, 14, 615–623, doi:10.1097/HJR.0b013e3280ecfd56.
- Rosenbaum, S.; Morell, R.; Abdel-Baki, A.; Ahmadpanah, M.; Anilkumar, T. v.; Baie, L.; Bauman, A.; Bender, S.; Boyan Han, J.; Brand, S.; et al. Assessing Physical Activity in People with Mental Illness: 23-Country Reliability and Validity of the Simple Physical Activity Questionnaire (SIMPAQ). BMC Psychiatry 2020, 20, 108, doi:10.1186/s12888-020-2473-0.
* It should be clear to describe the statistical methods and remove repeated questions.
The statistical methods are described in section 2.6. We believe that by repeating the research questions in this section, it will make it easier for the reader to understand how the data were analysed for each of the research questions.
* Was the association between the independent (physical activity preferences, barriers, or motivating factors) and dependent variable (physical activity patterns or sedentary behaviour) analysed using linear regression?
No.
* Were the independent variables analysed using principal component analysis or factor analysis in the data analysis section?
No.
lines 208-212 – The paragraph should move to the participants of the Methods.
We disagree. How many people responded and how they responded are results of the study.
* Tables 1 and 2 might be combined into one.
We would rather not combine Tables 1 and 2 as this would disrupt the flow of the results section, for which our intention was to present the data for each research question in turn.
In Table 1, for the independent variable sample size, please clarify whether the percentage of some variables is combined because they are too small for the analysis, for example, ethnicity should be divided into white and non-white, like Table 3.
In Table 1, we simply seek to describe the characteristics of the study participants. For this purpose, we do not see any benefit to presenting ethnicity as just either white or non-white.
Dealing with missing data should be mentioned in the statistical analysis of the Methods.
The analysis was performed on available data only, the number of participants with missing physical activity data was less than 5% and missing sedentary behaviour was 10%. Hence, we anticipate the potential bias caused by missing physical activity data to be low, particularly in the context that this survey is a voluntary sample.
We did also investigate the characteristics of the participants with missing physical activity data and there were no clear differences between them and the participants with complete physical activity data. We have added the following statement to the statistical analysis methods section; ‘All analyses were performed using available data, differences in characteristics between participants with and without completed physical activity were compared’.
In Table 2, there is a lack of time spent in total physical activity and need to recalculate levels of physical activity: vigorous, moderate, and light. It is not true because almost participants belong to vigorous or moderate physical activity with/without walking.
We did not seek to quantify time spent in total physical activity or light physical activity in this study. We wonder if the reviewer has misinterpreted the balance of physical behaviours reported in Table 2 by not realising that moderate PA and vigorous PA data are presented in min/week and time spent in SB in min/day. We acknowledge using different units may have caused confusion so have edited the table to include moderate PA, vigorous PA and time spent walking presented in both minutes per week and minutes per day.
* In Table 3, I would expect the authors to analyse the relationships of walking (≥ 150, 30-149, and < 30 min/day) and sedentary behaviour (≥ 120 vs. < 120 min/d) with sociodemographic and health-related variables.
Please see our response to your similar comment above.
In footnote, it should be mentioned that logistic regression was adjusted for all other variables except for the variable in the regression analysis.
Thank you, we have added this clarification in Table 3, footnote d.
In group comparison, the authors should mention a reference group in physically active with/without walking and muscle strengthening activity separately.
We have defined the reference group for each of physically active with/without walking and muscle strengthening activity separately in the footnote of Table 3.
lines 238-276 – In the text, it’s better to present the results with OR and 95% CI in unadjusted analyses and show whether the significance of these variables disappeared or remained after adjustment. The unadjusted and adjusted associations should be shortened.
Given that we are not constrained by a word limit, we would prefer to keep with our more comprehensive approach to explaining these results.
* In Tables 4-6, physical activity preferences, barriers, and motivating factors should be analysed by principal component analysis or factor analysis, and then their relationship with physical activity, walking, sedentary behaviour, and muscle strengthening activity should be analysed using linear regression when controlling for sociodemographic and health-related variables. These tables should be combined into one.
We think it is important to include all the information that is currently presented in Tables 4-6 in the manuscript. Presenting the data for all preferences, barriers and motivating factors, enables us to answer research question 4 (What barriers to participating in physical activity are reported by people with SMI?) and 5 (What motivating factors for participating in physical activity are reported by people with SMI?); We think it is interesting and useful to the reader to see the raw survey responses to each of the individual questions.
We agree with the reviewer that principal component analysis would be a useful way of reducing the number of dimensions (factors) of the data and could then be used to investigate the relationship with participant physical activity preferences and other characteristics. However, we believe there is merit in comparing the response to each question separately so the difference in agreement between different characteristic groups can be investigated and presented individually.
We did consider including more formal analysis, but concluded that presenting descriptive statistics, alongside highlighting large differences, was most appropriate to answer research question 6. Particularly due to the observational nature of this research and the number of statistical tests that would need to have been performed.
* Figures 1-4 should move to the Supplement.
We would prefer to keep these figures in the manuscript because they provide a useful visual summary of key study data.
line 351 – In discussion, the first paragraph should be clarified. For example, the main finding that “large proportions of the sample were insufficiently active and excessively sedentary” was based on physical activity preferences, while in the results of Table 2, 76% of the participants had achieved more than 150 minutes of moderate and vigorous intensity activity a week.
43% of participants were insufficiently active when walking time was excluded from the calculation. We prefer to look at this outcome due to the potential for double counting of activity when walking time is included in the calculation. 26% of participants were classified as excessively sedentary and 68% did not meet the muscle strengthening recommendation of ≥2 sessions per week. Together, we think our interpretation is appropriate.
The discussion section should be considered to the issues relating to the methods, statistical analysis, and new results. Some clarifications of independent and dependent variables are needed to require for the results section and the author(s) needs to qualify some of their comments based on the weight of evidence provided by their statistical findings. The discussion does a reasonable job explaining the findings but unfortunately no explanatory mechanisms for the findings can be tested due to the limited statistical methods.
Considering our responses to your previously comments, we do not see any requirement to revise the discussion section. Our study did not seek to explore explanatory mechanisms and our statistical methods were appropriate for our research questions.
Reviewer 3 Report
To thank the authors for their scientific dedication to advancing knowledge of BP levels, lifestyle and BP preferences in people with SMI. The authors are encouraged to carry out BP interventions in patients with SMI to contribute to their quality of life.

Author Response
Comments and Suggestions for Authors:
To thank the authors for their scientific dedication to advancing knowledge of BP levels, lifestyle and BP preferences in people with SMI. The authors are encouraged to carry out BP interventions in patients with SMI to contribute to their quality of life.
Thank you for reviewing our manuscript.
Proposal for detailed modifications:
Title. The article focuses on SMI schizophrenia and depression, so the title should be changed. The following title is suggested: Exploring Patterns, Preferences, Barriers, and Motivating Factors for the practice the Physical Activity in People with schizophrenia and bipolar disorder
Thank you for this suggestion. We have updated the title to: “Physical Activity in Adults with Schizophrenia and Bipolar Disorder: A Large Cross-Sectional Survey Exploring Patterns, Preferences, Barriers, and Motivating Factors”. The revised title provides a clearer indication of the study population. We didn’t change the second part of the title because we thought it important for it to specify the study design.
Abstract
Specific comments
Question 1 - It is suggested that acronyms (abbreviations) should not be included in the Abstract. Eg.: SMI, PA. These abbreviations are indicated the first time the terms are referred in the Introduction.
We have removed all abbreviations from the abstract, as suggested.
Question 2 - The M and SD of the sample under study should be indicated. For example: 2446 teachers (X woman and X men) from ..., aged 30-55 years (M = 30.18; SD = 5.55), distributed in groups according to country, participated in the study.
The abstract already indicated the sample size (n=529). We have added the proportion of males and mean age.
Question 3 - It is recommended that the keywords are different from the exact words in the title. Eg.: sedentary, physical exercise, reasons for practise physical exercise, fitness level, severe mental illness
We have replaced ‘physical activity’ with ‘exercise’ and ‘sedentary behaviour’.
Introduction
Specific comments
Question 4 - The introduction should begin reflected the problem statement regarding the prevalence of people with severe mental illness who are sedentary and inactive.
Thank you. We are satisfied with the current content and flow of the introduction and do not see any value in making the proposed change.
Question 5 - A sedentary lifestyle is not the same as physical inactivity.
We make the distinction between sedentary behaviour and physical inactivity at the start of the second paragraph.
Question 6 - Next, it should be argued, described and specified what severe pathologies beyond schizophrenia and bipolar disorder are. To do this, the DSM-V manual of the American Psychiatric Association should be consulted. Once SMI has been clinically justified, the authors must highlight the scientific novelty of this study and the relationship with other clinical trial studies, intervention studies, correlational studies, etc. on BP in people with SMI.
We are satisfied that our study has been adequately introduced. The novelty of the work is highlighted in the discussion.
The meta-analysis to which the authors refer is from 2016, not recent. Citations from more recent studies are provided:
Schuch, F., Vancampfort, D., Firth, J., Rosenbaum, S., Ward, P., Reichert, T., Bagatini, N. C., Bgeginski, R., & Stubbs, B. (2017). Physical activity and sedentary behavior in people with major depressive disorder: A systematic review and meta-analysis. Journal of affective disorders, 210, 139–150. https://doi.org/10.1016/j.jad.2016.10.050
Vancampfort, D., Firth, J., Schuch, F. B., Rosenbaum, S., Mugisha, J., Hallgren, M., Probst, M., Ward, P. B., Gaughran, F., De Hert, M., Carvalho, A. F., & Stubbs, B. (2017). Sedentary behavior and physical activity levels in people with schizophrenia, bipolar disorder and major depressive disorder: a global systematic review and meta-analysis. World psychiatry: official journal of the World Psychiatric Association (WPA), 16(3), 308–315. https://doi.org/10.1002/wps.20458
O'Donoghue, B., Mifsud, N., Castagnini, E., Langstone, A., Thompson, A., Killackey, E., & McGorry, P. (2022). A single-blind, randomised controlled trial of a physical health nurse intervention to prevent weight gain and metabolic complications in first episode psychosis: the Physical Health Assistance in Early Psychosis (PHAstER) study. BJPsych open, 8(6), e189. https://doi.org/10.1192/bjo.2022.590
Rocks, T., Teasdale, S. B., Fehily, C., Young, C., Howland, G., Kelly, B., Dawson, S., Jacka, F., Dunbar, J. A., & O'Neil, A. (2022). Effectiveness of nutrition and dietary interventions for people with serious mental illness: systematic review and meta-analysis. The Medical journal of Australia, 217 Suppl 7(Suppl 7), S7–S21. https://doi.org/10.5694/mja2.51680
The citations we provide in the introduction are all relevant to the points we make. We cite the paper by Vancampfort et al. (2017) in the discussion.
Question 7 - It is also recommended to delete the text from lines 67 to 78 and to include the hypotheses of the present research study. At the end of the Introduction, the objectives of the study and the hypotheses put forward in relation to the scientific literature reviewed should be set out. The descriptive data can be specified in the data analysis as answering these questions.
Our protocol did not include testing any specific hypotheses so it would be inappropriate to include new hypotheses at this stage.
Materials and Methods
General comments
Question 8 - The methodological part of a study that follows the scientific method begins with: a) "Research design"(design and type of methodology used); b) description of the "Participants" (N, age of N, socio-demographic characteristics); c) inclusion criteria, exclusion criteria, sampling techniques used; d) description of the Measures or instruments; e) Procedure; f) Intervention design and g) Data analysis. The development of this section is fundamental to ensure an adequate research design and development of the intervention based on the correct application of the scientific method.
Our write-up of this section conforms with the Journal’s guidelines for authors and describes things with sufficient detail to allow others to replicate. This section covers all the items which you have listed, except ‘intervention design’ which is irrelevant because this is a cross-sectional, non-interventional study.
Specific comments
Study Design
Question 9 - The research design used must be described, for example: Retrospective descriptive observational retrospective study with cross-sectional quantitative (Hernández Sampieri et al., 2014) and qualitative (data collection through semi-structured interview) methodology.
We explained that this is a cross-sectional study that used a survey.
Question 10 - The use of qualitative methodology is not clear from the wording of the measures. This would need to be clarified in the Measures section.
This study didn’t involve qualitative methods.
Note: another important recommendation for a good research design and therefore a good writing of the scientific report (study) is to follow the CONSORT criteria
(https://www.equatornetwork.org/reportingguidelines/consort/).
CONSORT is a guideline for reporting parallel group randomised trials. As such, it is not relevant to this cross-sectional study. That said, some of the CONSORT criteria are relevant to our study design, and these have been addressed in sufficient detail.
Participants
Question 11 - The information provided table 1 “Participant characteristics (n = 529)” (page 6) corresponds to the "Participants". In the Participants section, age, demographic characteristics, level of education, Self-rated mental health, percentage of men and women, etc. should be explained. In addition, the criteria for inclusion and exclusion of the sample.
The characteristics of the participants are summarised in the results section. This approach is consistent with the CONSORT criteria. There are not any further eligibility criteria to specify.
Question 12 - The statistical technique used to calculate the sample size (use of the G* power statistical package) should be provided to check whether the study has adequate prior statistical power.
As specified on lines 108-109, the target sample size was 500 participants based on the anticipated response rate and previous research. In the limitations section we say “… the study was exploratory in nature; we did not include statistical hypothesis testing, and instead present raw proportions, odds ratios and confidence intervals.”
Question 13 - It is suggested that a bibliographic citation be provided to justify the type of design developed according to: the N, the sampling technique, the characteristics of the sample, the methods used for the collection of information.
We re-checked this section and are satisfied that appropriate supporting references have been provided in all places where they are required.
Question 14- The sample universe is not specified and how it was reduced to a total sample of 529 participants. For example:
Image retrieved from article: Cantisano, L. M., Gonzalez-Soltero, R., Blanco-Fernández, A., & Belando-Pedreño, N. (2022). ePSICONUT: An e-Health Programme to Improve Emotional Health and Lifestyle in University Students. International journal of environmental research and public health, 19(15), 9253. https://doi.org/10.3390/ijerph19159253
Unfortunately, we don’t have figures for the ‘sample universe’ as it is not as clear cut as recruiting from a set group of people such as a group of students. The number of people with SMI in the recruiting NHS trusts is not a static figure as people will drop in and out of services throughout the duration of the recruitment period. In addition, it would not be possible to invite everyone with an SMI diagnosis within the trust to take part as we are reliant on clinicians screening caseloads and inviting potential participants and not all clinicians are in a position to be able to do this.
Question 15 - Include table of inclusion and exclusion criteria. For example:
Information retrieved from article: Cantisano, L. M., Gonzalez-Soltero, R., Blanco-Fernández, A., & BelandoPedreño, N. (2022). ePSICONUT: An e-Health Programme to Improve Emotional Health and Lifestyle in University Students. International journal of environmental research and public health, 19(15), 9253.
https://doi.org/10.3390/ijerph19159253
The eligibility criteria have already been specified in full.
Messures o Instruments
Question 16 - This section should be organised according to:
1) Differentiate the dependent and independent variables under study and differentiate between Socio-demographic variables, health and anthropometric variables, psychological variables, lifestyle variables…
2) Describe the quantitative and qualitative data collection instruments used in the study;
3) Of the quantitative instruments, it should be specified that an ad hoc questionnaire was designed for the collection of data on:
- Socio-demographic variables;
- health and anthropometric variables: smoking behaviour, consumption of fruit and
vegetables, and height and weight, self-rated mental health…
- lifestyle variables: physical activity and sedentary behaviours (physical activity
preferences, barriers, and motivating);
As mentioned above, we are satisfied with the content and low of the methods section. It conforms with the Journal’s guidelines for authors and describes things with sufficient detail to allow others to replicate. The points you list here have all been sufficiently covered.
Question 17 - The validation method (reliability and internal consistency) of the
questionnaire used is not clear. Also, if it is a questionnaire created for this study, the
statistical analyses should be presented, for example: indicates the criterion validity of the instrument and the authors who validated the instrument from which the items indicated are taken.
The survey included a range of tools and questions that have been used in studies previously. All components are described in the methods with supporting references. For physical activity and sedentary behaviour, references 15 and 17 are pertinent to your comment:
- Orrell, A.; Doherty, P.; Miles, J.; Lewin, R. Development and Validation of a Very Brief Questionnaire Measure of Physical Activity in Adults with Coronary Heart Disease. European Journal of Cardiovascular Prevention & Rehabilita-tion 2007, 14, 615–623, doi:10.1097/HJR.0b013e3280ecfd56.
- Rosenbaum, S.; Morell, R.; Abdel-Baki, A.; Ahmadpanah, M.; Anilkumar, T. v.; Baie, L.; Bauman, A.; Bender, S.; Boyan Han, J.; Brand, S.; et al. Assessing Physical Activity in People with Mental Illness: 23-Country Reliability and Validity of the Simple Physical Activity Questionnaire (SIMPAQ). BMC Psychiatry 2020, 20, 108, doi:10.1186/s12888-020-2473-0.
Question 18- The Cronbach's alpha or Mcdonal's Omega statistic should be included to determine the internal consistency of the questionnaires and scales used.
As in our answer to question 17, we included some questions and scales in the questionnaire that have been used in previous research and where we have done so we have described in the methods with supporting references. Validating the outcomes and questions was not an aim of our research. We are not aiming to use multiple questions to measure the same construct, and so we don’t think it is necessary to determine the internal consistency of any items of our questionnaire.
Procedure
Question 19 - This section is not described in the study. The procedure should detail the steps undertaken to carry out the study in terms of: when approval was obtained from the ethics committee and informed consents for the voluntary and anonymous participation of the sample. When and how the objective of the research was explained to the participants who make up the sample universe of the study.
These points are covered in section 2.4 – recruitment and procedures.
Data analysis
Question 20 – Revised the description for simplificated. For example: Descriptive statistics and pre-post statistically differences were calculated for the variables under study: the ones related to lifestyle (e.g. diet quality and physical exercise) and the ones related to the psychological status of the participants (e.g. subjective psychological well-being, anxiety and depression). The normality test was applied to the variables using the Shapiro-Wilk statistics. Based on a confidence level of 95% and an α error of 5% (.05), the null hypothesis (that the distribution was normal) was rejected, p-value ≤ .05. The reliability of the psychometric questionnaires was calculated through Cronbach's Alpha and McDonald's Omega statistic.
We are satisfied with our description of the data analysis procedures.
Results
General comments
In the Results section, descriptive data, percentages or frequencies and regression analysis should be included.
We agree and they already are.
Specific comments
Question 21 – Table 1 corresponds to the Participants section.
Important clarification: The format of the tables does not comply with APA 7th edition. View: https://owl.purdue.edu/owl/research_and_citation/apa_style/apa_formatting_and_style_guide/apa_tables_and_figures.html
The format of the tables conforms with the Journal’s guidelines for authors, which do not specify to use the APA 7th approach.
Question 22 – It is important that the tables are organised according to the statistical tests indicated in the data analysis. First, descriptive data on the variables under study should be presented:
Patterns of physical activity and sedentary behaviour in people with SMI; Physical activity preferences of people with SMI; Barriers to participating in physical activity are reported by people with SMI; Motivating factors for participating in physical activity are reported by people with SMI
We are satisfied with the content and flow of the results section, which has been organised into sub-sections that correspond with the research questions.
Question 23 –Secondly, the results of the univariate and multivariate regression test are to be presented: Activity preferences, barriers and motivating factors differ by gender, age group, physical activity status, and self-rated mental health in this population Important clarification: Simple and multiple regression analysis data are not correctly presented. It is important to revise this article: “Análisis de regresión lineal múltiple con SPSS: un ejemplo práctico” (file:///Users/noeliabelando/Downloads/22704-
Text%20de%20l'article-61419-3-10-20190701.pdf)
See previous response.
Question 24 - Important: An excess of graphs with poor image quality is observed when the data have been reflected in tables 4, 5 and 6.
We would prefer to keep the figures in the manuscript because they provide a useful visual summary of key study data. We have resubmitted the figures and ensured the meet the criteria for high resolution images provided by the journal (minimum 1000 pixels width/height, or a resolution of 300 dpi or higher).
Question 25 - Each table must include a table footer. For example: The description of the abbreviation of the statistics: OR (odds ratio), CI (confidence interval) is not given in table 3.
We have added the missing abbreviations to the relevant table footers.
Question 26 - Check the correct formatting of tables in terms of exceeding the margins of the document format according to the journal's guidelines.
We believe that our manuscript conforms to the Journal’s formatting guidelines. If the tables exceed the permitted margins, we would appreciate it if an Editor could point us towards the specific changes that are required.
Discussion
General comments
Question 27 - It is recommended that after the reminder about the objective of the study, the fulfilment or non-fulfilment of the hypotheses proposed be indicated.
As indicated above, the study did not test any hypotheses. The key findings have been described, covering all our research questions.
Specific comments
Question 28 - The first paragraph of the discussion (lines 351 to 366) corresponds to the "Conclusion" section.
It is common to highlight the key findings at the start of the discussion.
Conclusions
General comments
Question 29 - It is important that the description of the Conclusions be based on the objectives set and on whether or not they were achieved.
The conclusion covers all our research questions and highlights the implications for intervention developers.
Specific comments
Question 30 - The text of the conclusions is a reflection/discussion of the authors, not the conclusions of the results. It is necessary that the conclusions respond to each of the objectives stated at the end of the Introduction.
We disagree. The conclusions relate to our research questions.
References
Question 31 - Is import adapt the wording of the references to the journal's regulations.
We used the Journal’s recommended style guide when preparing the references. Reviewer 4 acknowledges that the references are written in accordance with the Journal’s guidelines.
Question 32 - Ensure that all citations in the text have been referenced and in the correct order of appearance.
We have done this.
Reviewer 4 Report
Thank you for the opportunity to review the manuscript.
The work submitted for review examines a topic of great relevance in the field of psychology. The topic is very important, and the conceptual analysis made in the text is quite deep. The literature consulted is quite current but the sample isn´t quite large (which is a limitation for your work). I would like to thank the efforts by the authors of the manuscript and congratulate them on the work. Overall, the writing is clear, the goals are well described, the introduction should explain the objectives of the study based on the review of the previous literature and the conclusions are properly made and presented. I consider that the constructs proposed in the abstract of the work are quite well explained. Therefore, the manuscript brings significant knowledge of the scientific literature so and still covers existing gaps in the field. On a formal level, the manuscript complies with the requirements of the Journal and references are written in accordance with the regulations of the Journal. The work is ambitious, and the results confirm most of the hypotheses and the relevance and potential of the work is therefore recognized, but this Reviewer considers that several changes are needed to the manuscript is publishable. In this sense, it should better explain the novelty and relevance of the work considering the previous empirical evidence and should better describe the practical implications. The process for selecting participants and the procedure should be better described. The study hypotheses should also be better explained. It should better describe the measuring instruments. On the other hand, the limitations of the study should be better explained. It should describe the discussion and conclusions of the work better and, above all, update the manuscript references (most should be from the last 5 years). Finally, I wish the Authors the best in continuing this line of research.
Best wishes for Authors.
Author Response
Thank you for the opportunity to review the manuscript. The work submitted for review examines a topic of great relevance in the field of psychology. The topic is very important, and the conceptual analysis made in the text is quite deep. The literature consulted is quite current but the sample isn´t quite large (which is a limitation for your work). I would like to thank the efforts by the authors of the manuscript and congratulate them on the work. Overall, the writing is clear, the goals are well described, the introduction should explain the objectives of the study based on the review of the previous literature and the conclusions are properly made and presented. I consider that the constructs proposed in the abstract of the work are quite well explained. Therefore, the manuscript brings significant knowledge of the scientific literature so and still covers existing gaps in the field. On a formal level, the manuscript complies with the requirements of the Journal and references are written in accordance with the regulations of the Journal. The work is ambitious, and the results confirm most of the hypotheses and the relevance and potential of the work is therefore recognized, but this Reviewer considers that several changes are needed to the manuscript is publishable.
Thank you for reviewing our manuscript. We are glad that you recognise the value of our work.
In this sense, it should better explain the novelty and relevance of the work considering the previous empirical evidence and should better describe the practical implications.
We have attempted to highlight the main novel aspects and practical implications of our work in the discussion. Having reviewed this content, we are satisfied with the current wording.
The process for selecting participants and the procedure should be better described.
We have described the recruitment process and study procedure in full. It is unclear what part(s) you think have been inadequately described.
The study hypotheses should also be better explained.
Our protocol did not include testing any specific hypotheses so it would be inappropriate to include new hypotheses at this stage.
It should better describe the measuring instruments.
We have described all components of the survey. It is unclear what part(s) you think have been inadequately described.
On the other hand, the limitations of the study should be better explained.
We have described all key limitations and note that Reviewer 1 said “Limitations are generous and describe appropriate biases.”
It should describe the discussion and conclusions of the work better and, above all, update the manuscript references (most should be from the last 5 years).
It is unclear what aspect of the discussion and conclusions you think have been inadequately described. The Journal’s guidelines do not require the use of references from within a specific publication date-range. All our current references are relevant to the manuscript.
Finally, I wish the Authors the best in continuing this line of research. Best wishes for Authors.
Thank you.
Round 2
Reviewer 2 Report
Excellent paper. Thank you all for your contribution.
Author Response
Excellent paper. Thank you all for your contribution.
Thank you for reviewing our manuscript.
Reviewer 3 Report
We would like to thank the authors for their efforts to improve the concreteness of the summary and the participants. However, it is important to address comments that have not been modified in the manuscript.
The following comments have not been modified as suggested in the following comments:
Question 4 - The introduction should begin by reflecting the problem statement regarding the prevalence of people with severe mental illness who are sedentary and inactive.
Question 6 - Next, you should argue, describe and specify which are the serious pathologies beyond schizophrenia and bipolar disorder. The DSM-V manual of the American Psychiatric Association should be consulted for this purpose. Once clinically justified, the authors should highlight the scientific novelty of this study and the relationship to other clinical trial studies, intervention studies, correlational studies, etc. on BP in people with SMD.
- One of the key aspects of the correct application of the scientific method is the formulation of hypotheses.
- Citations to more recent studies are not included.
Methodology
The design of the study and the preparation of the ad hoc survey are not specified, nor is an explanation of the data analysis for each of the results in accordance with the objectives set out.
Results
Simplify the tables and graphs according to the objectives of the study.
Author Response
Reviewer 3:
We would like to thank the authors for their efforts to improve the concreteness of the summary and the participants. However, it is important to address comments that have not been modified in the manuscript.
Thank you for reviewing our manuscript again.
The following comments have not been modified as suggested in the following comments:
Question 4 - The introduction should begin by reflecting the problem statement regarding the prevalence of people with severe mental illness who are sedentary and inactive.
We disagree. The first paragraph of the introduction highlights the mortality gap experienced by people with SMI. This is the problem that our programme of research in physical activity and SMI seeks to address, and as such is a relevant starting point for our paper.
Question 6 - Next, you should argue, describe and specify which are the serious pathologies beyond schizophrenia and bipolar disorder. The DSM-V manual of the American Psychiatric Association should be consulted for this purpose. Once clinically justified, the authors should highlight the scientific novelty of this study and the relationship to other clinical trial studies, intervention studies, correlational studies, etc. on BP in people with SMD.
We are satisfied with the current flow of the introduction section. It provides sufficient background and justification to the study. Novel features are highlighted in the discussion.
- One of the key aspects of the correct application of the scientific method is the formulation of hypotheses.
This was an exploratory study in which we collected data to see what patterns might emerge. We did not have a specific hypothesis in mind. One might think of it as hypothesis-generating research rather than confirmatory or hypothesis-testing research.
- Citations to more recent studies are not included.
We were satisfied that the references we had used were appropriate for supporting the points we made in our manuscript. None of them are out-of-date or invalid. We note that the fourth reviewer said that our references are “quite current”.
Methodology
The design of the study and the preparation of the ad hoc survey are not specified, nor is an explanation of the data analysis for each of the results in accordance with the objectives set out.
The study design is specified in section 2.2. The components of the survey are explained in section 2.5, in which we also include a link to a full version of the survey. In section 2.6, we describe the data analysis procedures for each of our research questions.
Results
Simplify the tables and graphs according to the objectives of the study.
The manuscript contains a lot of data, which we have tried to present in a clear and comprehensive format. The other three reviewers seem happy with how we have presented the data, and we would prefer to leave the tables and figures as they are.
Reviewer 4 Report
Thank you for the opportunity to review the manuscript again. Overall, the writing is clear, the goals are well described, well-considered introduction and the results properly made and presented. Therefore, the manuscript brings significant knowledge of the scientific literature so and still covers existing gaps in the field of Education. Therefore, my assessment is positive for the publication of this work, with a new suggestion.
Firstly, I would like to thank the efforts by the authors of the manuscript and congratulate them on the work. I recognize that they have considered almost all considerations of the Reviewers. Clearly, all the comments from Reviewers have contributed to a better quality of the manuscript. I have checked in the revised manuscript are corrected the most of errors found by the reviewers, both formally and content.
Secondly, I have verified that the information is presented in a clear and organized way in subtitles. It assumes good work with great potential.
Thirdly, in the Discussion section appears practical and educational implications and future directions correctly described. I have found the manuscript show a paragraph of study limitations. However, the conclusions section is somewhat concise. It is precisely a section to highlight the main practical implications of the study.
Fourthly, I have found that all the references are correctly written. The references are quite current, and all references comply with the Journal style.
Fifthly, I have verified that the format of the figures complies with the regulations of the Journal.
Finally, considering the changes made to the manuscript by the authors and the new suggestion, I consider that the manuscript can continue with the review process, considering the opinion and suggestions of other Reviewers.
Best wishes for Authors.
Author Response
Thank you for the opportunity to review the manuscript again. Overall, the writing is clear, the goals are well described, well-considered introduction and the results properly made and presented. Therefore, the manuscript brings significant knowledge of the scientific literature so and still covers existing gaps in the field of Education. Therefore, my assessment is positive for the publication of this work, with a new suggestion.
Thank you for reviewing our manuscript again.
Firstly, I would like to thank the efforts by the authors of the manuscript and congratulate them on the work. I recognize that they have considered almost all considerations of the Reviewers. Clearly, all the comments from Reviewers have contributed to a better quality of the manuscript. I have checked in the revised manuscript are corrected the most of errors found by the reviewers, both formally and content.
Thank you.
Secondly, I have verified that the information is presented in a clear and organized way in subtitles. It assumes good work with great potential.
Thank you.
Thirdly, in the Discussion section appears practical and educational implications and future directions correctly described. I have found the manuscript show a paragraph of study limitations. However, the conclusions section is somewhat concise. It is precisely a section to highlight the main practical implications of the study.
Thank you. We are content that the following sentence from the conclusions section highlights the main practical implication of the study: “The information from this study has the potential to guide physical activity intervention planning and programming, which could contribute to decreasing the chronic disease burden, and improving psychological wellbeing in adults with SMI.”
Fourthly, I have found that all the references are correctly written. The references are quite current, and all references comply with the Journal style.
Thank you.
Fifthly, I have verified that the format of the figures complies with the regulations of the Journal.
Thank you.
Finally, considering the changes made to the manuscript by the authors and the new suggestion, I consider that the manuscript can continue with the review process, considering the opinion and suggestions of other Reviewers.
Thank you.